# Inhibition of protein arginine deiminase 4 prevents inflammation-mediated heart failure in arthritis

Lukas A Heger[1,2,3], Nicolas Schommer[1,3], Shoichi Fukui[1,2], Stijn Van Bruggen[1,4], Casey E Sheehy[1], Long Chu[1], Sridharan Rajagopal[5], Dhanalakshmi Sivanandhan[5], Bruce Ewenstein[1,2], Denisa D Wagner[1,2,6]

Rheumatoid arthritis is a prototypic inflammatory condition with affected patients being at greater risk of incident heart failure (HF). Targeting innate immune cell function in the pathogenesis of HF bears the potential to guide the development of future therapies. A collagen-induced arthritis (CIA) model in DBA/1 J mice was used to generate arthritis. Mice with CIA developed concentric hypertrophic myocardial remodeling, left ventricular (LV) diastolic dysfunction, and HF with elevated plasma B-type natriuretic peptide levels but preserved LV ejection fraction. Key features of HF in CIA were increased infiltration of activated neutrophils, deposition of neutrophil extracellular traps in the myocardium, and increased tissue levels of the proinflammatory cytokine IL-1β. Specific inhibition of protein arginine deiminase 4 (PAD4) by an orally available inhibitor (JBI-589), administered after the onset of clinical arthritis, prevented HF with reduced neutrophil infiltration. We identify PAD4-mediated neutrophil activation and recruitment as the key thromboinflammatory pathway driving HF development in arthritis. Targeting PAD4 may be a viable therapeutic approach for the prevention of HF secondary to chronic inflammation.

## Introduction

The activity of protein arginine deiminase 4 (PAD4) is comprehensively involved in the pathophysiology of rheumatoid arthritis (RA) partly through the production of immunogenic neoepitopes (1). Accordingly, altered PAD4 activity is amongst the early indicators preceding the onset of RA (2). We have shown that prophylactic treatment with the non-covalent PAD4 inhibitor JBI-589 alleviates arthritis in a mouse model (3). The effect of therapeutic PAD4 inhibition on disease severity and the extent of the incidental collateral damage after onset of arthritis, however, it is not defined. In neutrophils, activation and nuclear translocation of PAD4 facilitates histone modification, an essential step in the process of the formation of neutrophil

extracellular traps (NETs) (4, 5, 6). Accumulating evidence points to NETs as an important risk factor for thrombosis with associated inflammation (thromboinflammation) in arthritis (7, 8). The clinical relevance of aberrant thromboinflammation in arthritis is supported by an increased risk of adverse myocardial remodeling and heart failure (HF) among affected patients (9). Current anti-inflammatory RA therapies, although effective in improving joint disease severity, appear to have little to no cardioprotective effect, highlighting the need for the development of better targeted therapies (10, 11). We have shown, using a murine acute myocardial infarction/reperfusion injury model, that targeting early leukocyte recruitment and formation of NETs offers significant protection against myocardial damage (12). Indeed, evidence indicates, that immune cell infiltration in the myocardium can have adverse effects on the heart and contribute to the pathogenesis of HF in a chronic setting as well (13). We hypothesized that targeting PAD4 could break a vicious cycle of NETs triggering the release of von Willebrand factor (VWF) and P-selectin from Weibel-Palade bodies, which in turn facilitate increased leukocyte recruitment. Together, with the ability of extracellular PAD4 to reduce VWF-platelet string clearance and accelerate the formation of stable platelet plugs; this could provide the basis for aberrant thromboinflammation (14).

In addition to its enzymatic role in chromatin decondensation and NET release, PAD4 is also critically involved in neutrophil production of IL-1β, a biomarker of thromboinflammation, via regulation of the neutrophil-to-lymphocyte ratio (NLR) family pyrin domain containing 3 (NLRP3) inflammasome activity (15). Correspondingly, in a model of murine peritonitis, the absence of NLRP3 inflammasome activity also affects neutrophil recruitment (16).

We have shown that deposition of VWF in the synovium and subsequent pathogenic NET retention promotes arthritis and established a critical role of PAD4 (7, 17).

We speculated that elevated PAD4 activity, as described in the well-established model of collagen-induced arthritis (CIA) in the DBA/1 strain, could drive systemic thromboinflammation and neutrophil activation and recruitment to the myocardium, thus mediating adverse myocardial remodeling in the late stages of the

[1]Program in Cellular and Molecular Medicine, Boston Children's Hospital, Boston, MA, USA [2]Department of Pediatrics, Harvard Medical School, Boston, MA, USA [3]Departement of Cardiology and Angiology, University Hospital Freiburg Bad Krozingen, Freiburg, Germany [4]Center of Molecular and Vascular Biology, Department of Cardiovascular Science, KU Leuven, Leuven, Belgium [5]Jubilant Therapeutics Inc., Bedminster, NJ, USA [6]Division of Hematology/Oncology, Boston Children's Hospital, Boston, MA, USA

Correspondence: denisa.wagner@childrens.harvard.edu

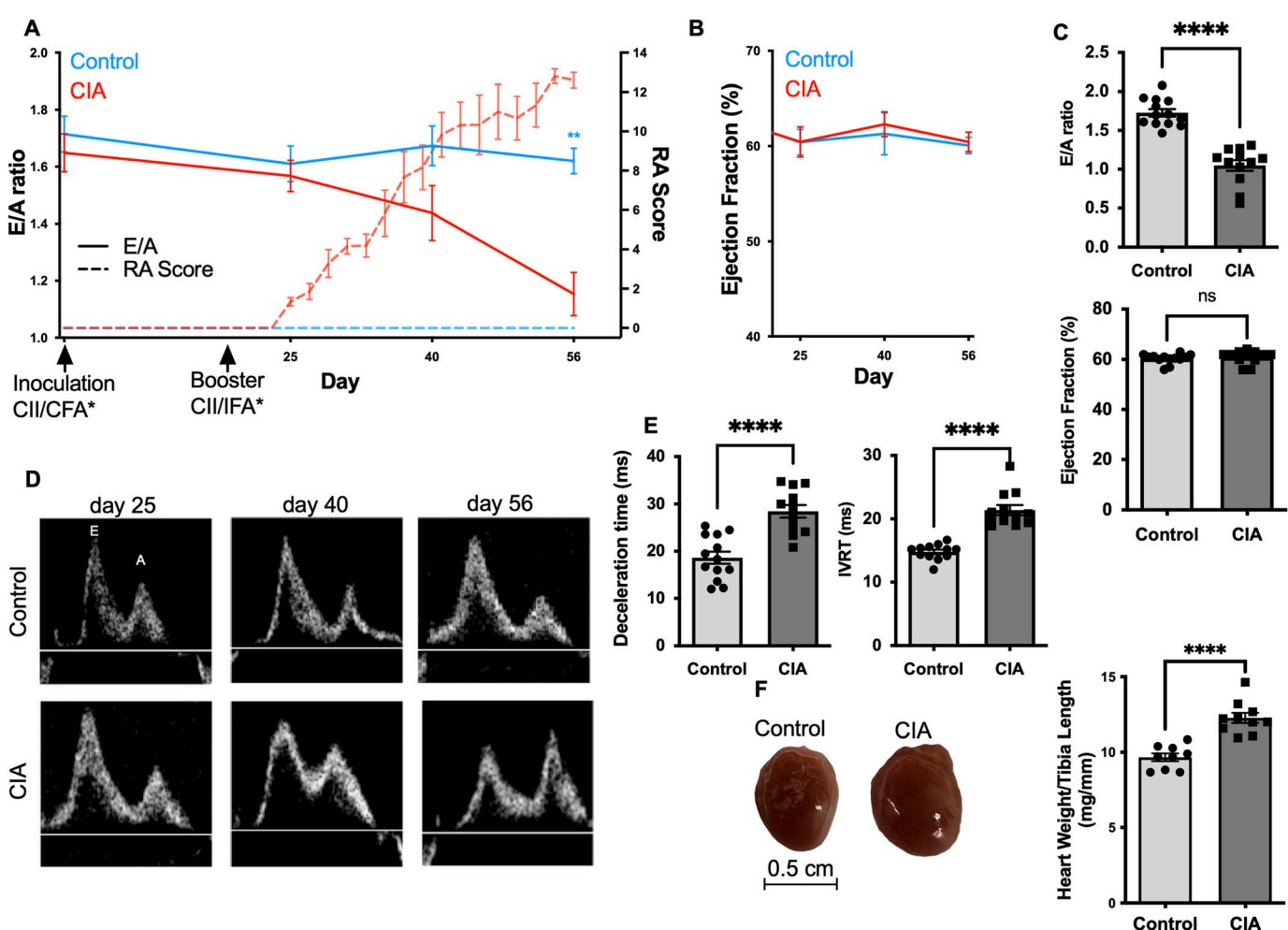

**Figure 1. Mice with collagen-induced arthritis (CIA) develop diastolic dysfunction and myocardial hypertrophy.**
**(A, E)** Time course illustrating the ratio of early (E) to late (A) diastolic filling velocities across the mitral valve as measured by pulsed-wave Doppler (E/A ratio; left Y axis; —). Clinical severity of rheumatoid arthritis (RA score; right y axis; —) in DBA/1 J mice with CIA and a healthy control, respectively. (n = 6) Arrows indicate the timepoint of inoculation/booster with type II collagen and (*) complete/incomplete Freud's adjuvant. **(B)** Left ventricular ejection fraction (EF) in DBA/1 J mice with CIA and a healthy control, respectively, demonstrated over the course of time (n = 6). **(C)** E/A ratio and EF as assessed by echocardiography in mice with CIA and healthy control at day 56 (n = 12). **(D)** Representative flow patterns, acquired using pulsed-wave Doppler echocardiography, depicting the velocities over the mitral valve at the indicated timepoints in mice with CIA and healthy control, respectively. **(E)** Deceleration time reflecting the amount of time needed to equalize the pressure difference between the left atrium (LA) and the left ventricle (LV) or the time interval from the peak of the E-wave to its projected baseline and isovolumetric relaxation time a marker for myocardial relaxation measuring the time for crossover between the LA and LV pressures as assessed by echocardiography in arthritic mice with CIA and healthy controls at day 56 (scale bar 0.5 cm) (n = 12). **(F)** Heart weight normalized to tibia length in CIA mice and the healthy control group after 56 d. (n = 9) Data are mean ± SEM. *P < 0.05, **P < 0.01, ***P < 0.001; unpaired t test (isovolumetric relaxation time: Mann–Whitney U test).

disease. Our effort to investigate the role of PAD4 was facilitated by the recent description of an orally available selective PAD4 inhibitor, JBI-589 (3).

## Results

### Mice with CIA develop diastolic dysfunction and myocardial hypertrophy

The CIA model in DBA/1 mice is the most widely used model to reproduce clinical symptoms of human inflammatory RA (Fig S1A and B) (18). In the present study, the DBA/1-strain mice were immunized two times with a bovine type II collagen (CII) emulsion in complete Freund's adjuvant on day 0 and then in incomplete Freund's adjuvant on day 21 (17). The first clinical signs of arthritis (ankle, paw swelling) became visible after the booster injection from day 25 (±2.3). After 8 wk, DBA/1 J mice developed an average arthritis severity of 12 (±1.9) (Fig 1). Echocardiography at this time showed that CIA resulted in significantly reduced diastolic peak early to late filling velocity (E/A ratio) when compared with the healthy control indicative of impaired filling pressures (1.1 ± 0.3 versus 1.7 ± 0.2; n = 12; P < 0.0001) with no difference in systolic left ventricular ejection fraction when compared with the healthy control mice (62% ± 4% versus 63% ± 4%; n = 12; P = 0.6) (Fig 1A–D). Both, mice with arthritis and healthy controls had equal

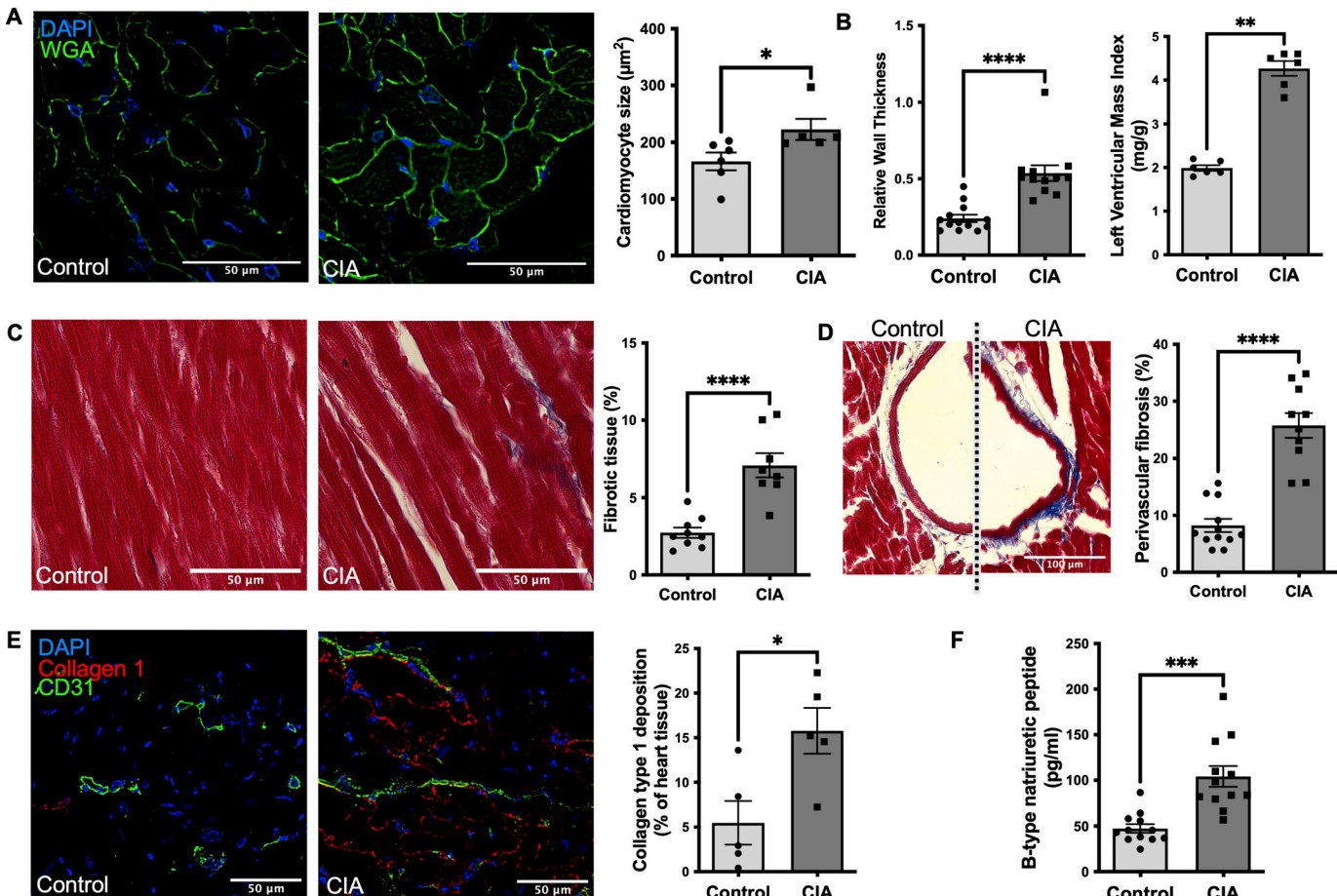

**Figure 2. Left ventricular remodeling and fibrosis are present in mice with collagen-induced arthritis (CIA) with clinically relevant diastolic dysfunction mimicking heart failure in RA patients.**
**(A)** Representative images of LV sections stained with WGA for cell membrane and DAPI staining of DNA to measure cardiomyocyte cell size defined by the cross-sectional area (Scale bar 50 $\mu$m) (n = 6). **(B)** Left ventricular mass, a parameter estimating total LV weight was calculated using Vevo LAB ultrasound analysis software as the difference between the epicardium-delimited volume and the LV chamber volume multiplied by an estimate of myocardial density. Diastolic LV posterior wall thickness (LVPWd) was measured at the end diastole as a measure of LV geometry and expansion (n = 12). **(C)** Representative LV section images of Masson's trichrome staining for fibrotic tissue (blue) in a mouse with CIA and a healthy control, respectively. Comparative analysis of the total area of fibrotic tissue in LV sections (Scale bar 50 $\mu$m) (n = 9/8). **(D)** Representative images of Masson's trichrome staining with perivascular fibrosis and quantitative analysis (scale bar: 100 $\mu$m). (For each mouse, five vessels chosen at random were quantified and their average was used for comparative analysis; n = 10). **(E)** Quantification of collagen type 1 (red) and vasculature (CD31$^+$ cells) in LV. Representative images are shown (scale bar: 50 $\mu$m) (n = 5). **(F)** Plasma levels of brain natriuretic peptide, a hormone produced by the body when the heart is enlarged, as measured by ELISA on day 56 in mice with CIA and a healthy control group (n = 12). Data are mean ± SEM. **(F)** *P < 0.05, **P < 0.01, ***P < 0.001; Mann–Whitney *U* test (F) *t* test.

heart rates at the time of the measurement (n = 12; *P* = 0.7). As impaired filling pressures are because of impaired left ventricular relaxation/stiffness, they are also reflected in changes to the cardiac cycle. Correspondingly, in mice with CIA, it took longer for the pressure in the LV to drop below that of the left atrium delaying the left ventricular filling process, as reflected in a prolonged isovolumetric relaxation time (IVRT) (19.1 [16.8–22.9] ms versus15.3 [14.3–15.8] ms; n = 12; *P* = 0,004). Similarly, the duration for equalizing the pressure difference between the left atrium and the left ventricle (E-wave deceleration time [DT]) was prolonged in mice with CIA when compared with the healthy control group (29 ± 5 ms versus 19 ± 4 ms; n = 12; *P* < 0.0001) (Fig 1E).

In addition to the impaired relaxation of the LV as determined by echocardiography, we observed that mice with CIA developed cardiac hypertrophy with markedly increased heart weight to tibia

length ratio (11.9 ± 0.7 mg/mm versus 9.7 ± 0.8 mg/mm; n = 12; *P* < 0.0001) (Fig 1F).

### Left hypertrophic ventricular remodeling and fibrosis are present in mice with CIA with clinically relevant diastolic dysfunction mimicking heart failure in RA

Hypertrophic myocardial remodeling in CIA mice was supported by an ~15% increase in cardiomyocyte cell size (n = 5–6; *P* = 0.03) and elevated measurements reflecting hypertrophic LV geometry such as an increased relative wall thickness (RWT) (0.45 [0.4–0.5] versus 0.28 [0.2–0.3]; n = 12; *P* = 0.004), a 75% increase in the end-diastolic left ventricular posterior wall thickness (LVPWd) (n = 12; *P* < 0.001) and an increased LV mass (60 [52–75] mg versus 40 [35–57] mg; n = 12 *P* = 0.04), when compared with a healthy age-matched control

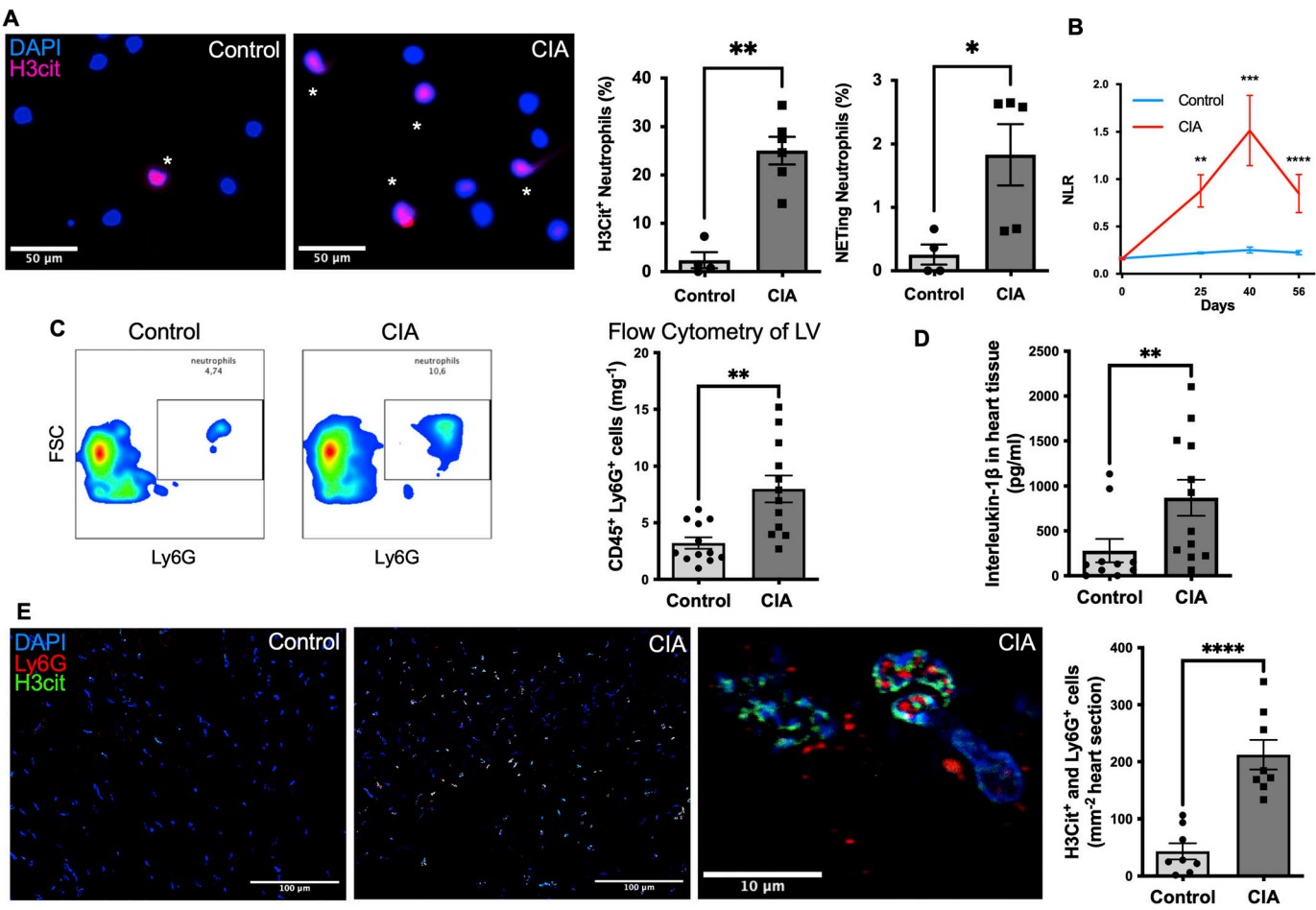

**Figure 3. Neutrophils are activated (showing histone citrullination) in the circulation and myocardium of mice with collagen-induced arthritis (CIA).**
**(A)** Representative immunofluorescence images of isolated neutrophils from blood of mice with CIA or healthy control mice. Neutrophils from mice with CIA are pre-activated (H3Cit$^+$; pink) and show a propensity for spontaneous NET formation. Representative microscopic picture of isolated neutrophils are shown (Scale bar 50 $\mu$m). Healthy control (left) and CIA (right). * = activated H3Cit$^+$ neutrophil. (n = 6). **(B)** Graph illustrating, as a function of time, the neutrophil-to-lymphocyte ratio a marker of hyperinflammatory response, measured in peripheral blood in DBA/1 J mice with CIA and a healthy control, respectively (n = 6). **(C)** Gating and quantification of infiltrating Ly6G$^+$/CD45$^+$ and CD45$^+$ cells using flow cytometry in healthy control and CIA myocardial tissue (n = 14). **(D)** Quantitative comparison of tissue levels of IL-1$\beta$ an inflammatory cytokine-mediating fibrosis measured using ELISA (n = 9). **(E)** Representative LV sections of CIA mice and healthy control mice stained for DAPI$^+$ (blue), Ly6G$^+$ (red), and H3Cit$^+$ (green) cells (scale bar: 100 $\mu$m) and a representative image of an activated neutrophil in the right panel with co-staining of Ly6G$^+$ (red) and H3Cit$^+$ (green) (scale bar 10 $\mu$m). Comparative analysis of the total number of Ly6G$^+$ (red) and H3cit$^+$ (green) cells in LV heart sections of CIA mice and healthy controls (n = 8). Data are mean ± SEM. **(C, E)** *$P$ < 0.05, **$P$ < 0.01, ***$P$ < 0.001; Mann–Whitney $U$ test (C, E) $t$ test.

group (Fig 2A and B). With increased RWT and left ventricular mass (LVM), the left ventricular hypertrophy in mice with CIA can be characterized as concentric. Histological analysis showed that myocardial hypertrophy was likely based on the increased size of cardiomyocytes and increased deposition of extracellular matrix (myocardial fibrosis). Mice with CIA showed increased myocardial fibrosis when compared with the healthy control both of the total heart sections (7.9% ± 3% versus 3.2% ± 1.8%; n = 8–9 $P$ = 0.001) and notably in the perivascular tissue (25.5% ± 7.3% versus 8.2% ± 3.5%; n = 10-12; $P$ < 0.0001) (Figs 2C and D and S1C). Immunofluorescent analysis showed elevated deposition of collagen type 1 in mice with CIA when compared with healthy controls (15 [11–21]% versus 3 [1–11]%; n = 5; $P$ = 0.03) in the LV (Fig 2E). Taking into account the inherent limitations in our experimental setup (swelling of paws in mice with CIA), to test the cardiac symptoms of HF such as breathlessness and fatigue during exercise, we relied on the

plasma levels of B-type natriuretic peptide (BNP), a clinically used biomarker of HF, to confirm the presence of HF as described before (19). Mice with CIA had significantly increased BNP levels when compared with healthy control mice (104 ± 19.9 pg/ml versus 47.4 ± 16.4 pg/ml; n = 12; $P$ < 0.0001) (Fig 2F). LV diastolic dysfunction with preserved ejection fraction and LV structural remodeling as seen in CIA are hallmarks of heart failure with preserved ejection fraction (HFpEF), a HF phenotype that is increasing in prevalence with ill-defined basic mechanisms (20, 21).

## Neutrophils are activated both in the circulation and myocardium of mice with CIA

We hypothesized that chronic neutrophil activation in CIA would be detrimental to the heart by promoting neutrophil infiltration

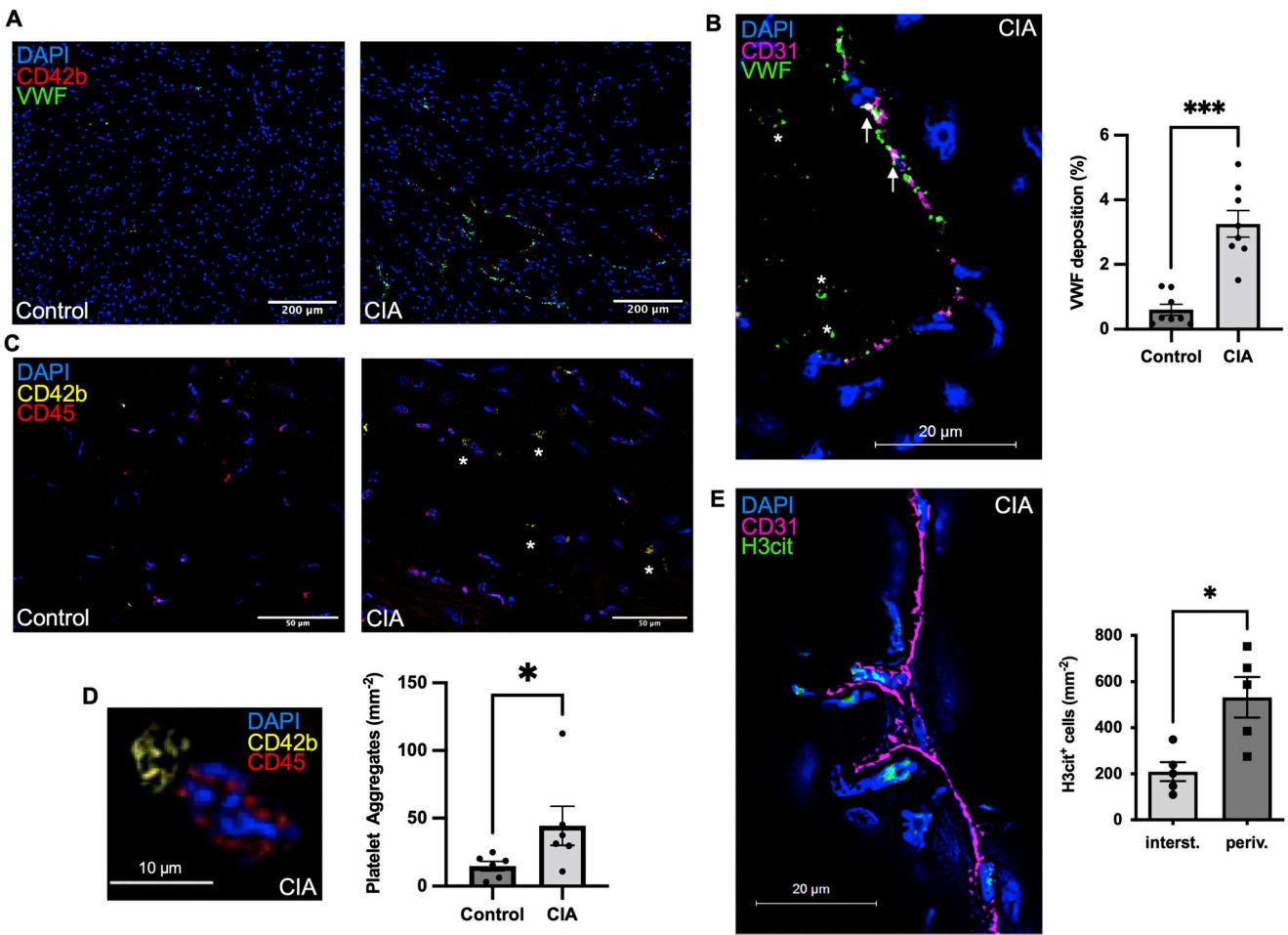

**Figure 4. Thromboinflammation in and around the cardiac microvessels is present in mice with Collagen-induced arthritis (CIA).**
**(A)** Representative LV sections and quantification of von Willebrand factor (VWF) staining, an endothelial adhesive protein deposited in the vasculature (scale bar: 200 μm) (n = 7). **(B)** Endothelial activation leads to extrusion of endothelial-anchored VWF within the vascular lumen. Arrows indicate VWF near the vessel wall and * indicates intraluminal VWF aggregates; L = lumen of vessel (scale bar: 20 μm). **(C)** Immunofluorescence staining in LV sections of mice with CIA and healthy control mice for accumulating CD42b⁺ cells indicative of (micro) thrombosis in the vasculature a major feature of thromboinflammation (Scale bar 50 μm). **(D)** Macroscopic picture of an aggregate of CD42b⁺ platelets in the myocardium of a mouse with CIA (scale bar: 10 μm) and quantification of total number of aggregates per LV section. (n = 5). **(E)** Immunofluorescence staining in LV sections of mice with CIA for H3Cit⁺ cells in relation to CD31⁺ signal with quantification of H3Cit⁺ cells. (For each mouse, five peripheral and five perivascular cross-section areas were chosen blindly, quantified, and the resulting total average used for comparative analysis [n = 5]) (scale bar: 20 μm). Data are mean ± SEM. *P < 0.05, **P < 0.01, ***P < 0.001; Mann–Whitney U test.

in the myocardium with subsequent NET formation. Indeed, as we predicted, neutrophils isolated from mice with CIA were more preactivated/H3Cit⁺-positive (49 [34–62]% H3Cit⁺cells versus 1 [0–6]% H3Cit⁺cells; n = 4–6; P = 0.009) with a higher propensity for NETosis when compared with neutrophils isolated from blood of healthy controls (3 [0.7–3]% of cells versus 0.2 [0–0.6]% of cells; n = 4–6; P = 0.03) (Fig 3) and higher plasma levels of NETosis markers H3Cit and double-stranded desoxyribonucleic acid at the indicated timepoints (Fig S1D and E). The chronic proinflammatory setting of arthritis was reflected by a higher neutrophil-to-lymphocyte ratio at the indicated timepoints (Fig 3B) throughout the 8-wk experiment in the DBA/1 J mice with CIA when compared with healthy controls (n = 6; Day 25: P = 0.002/Day 40: P = 0.002/Day 56: P = 0.004).

Indeed, as we predicted, FC analysis of heart tissue lysates showed higher counts of neutrophils in the myocardium of mice with CIA when compared with a healthy control group (8 ± 4 cells/mg per LV tissue versus 3 ± 2 cells/mg per LV tissue: n = 12; P = 0.001) (Fig 3C) also when compared with the total number of infiltrating CD45⁺ cells (n = 12; P = 0.001). Interestingly, there was no difference in the number of infiltrating CD45⁺ cells (n = 12; P = 0.4) (Fig 3C).

To check for activated neutrophils in the process of NETosis in heart tissue, we performed immunofluorescence analysis of heart sections of the LV searching for H3Cit⁺ and Ly6G⁺ double-positive cells. Indeed, mice with CIA showed a five-times higher number of H3Cit⁺ neutrophils when compared with healthy controls (212 ± 73 double⁺ cells/mm² versus 43 ± 40 double⁺ cells/mm²; n = 8 P < 0.0001) (Fig 3E). Corresponding with the higher levels of activated neutrophils, an IL-1β ELISA of heart tissue lysate of mice with CIA showed also higher values of IL-1β (1,325 ± 891.1 pg/ml versus 549.8

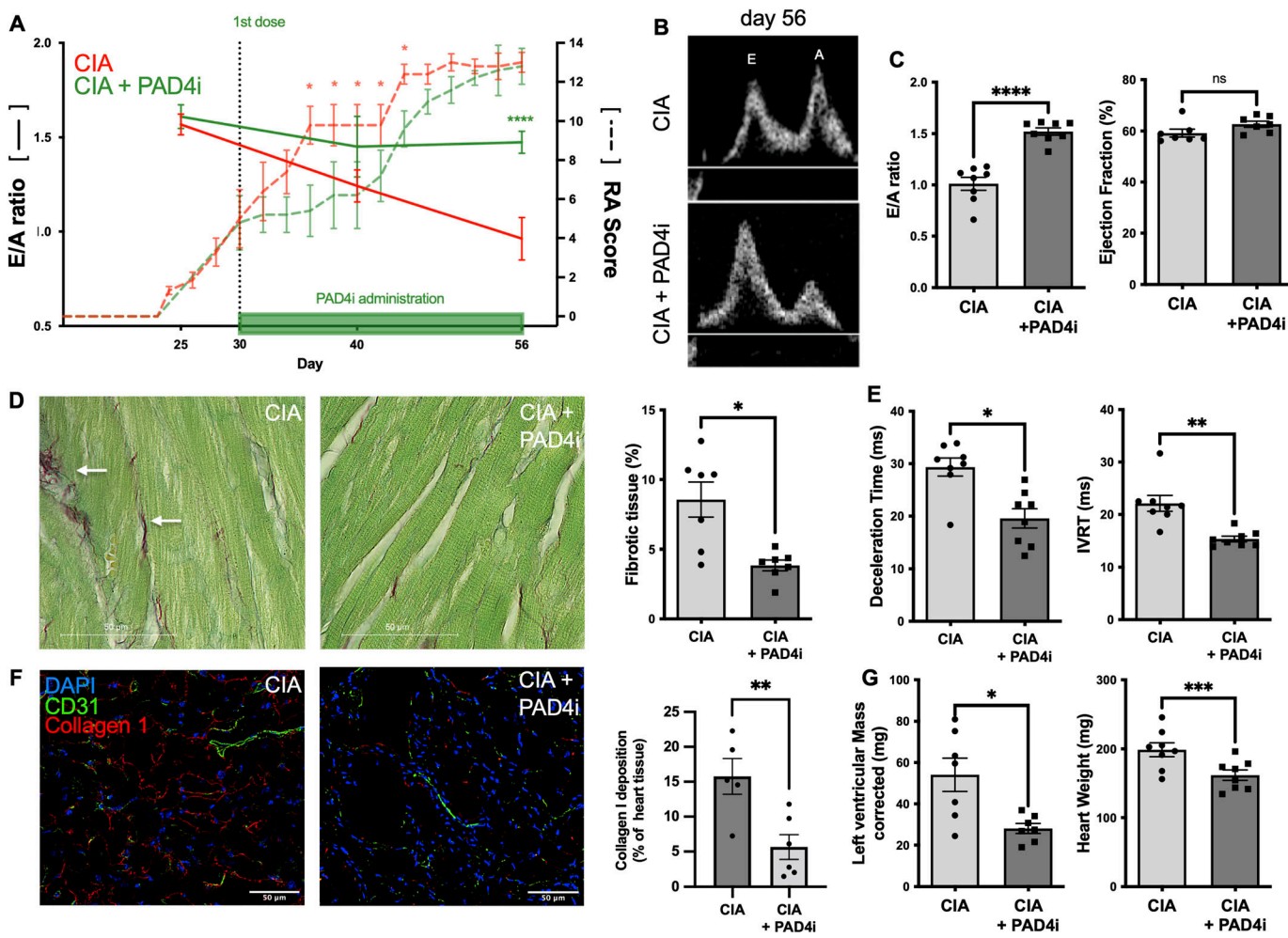

**Figure 5. Treatment with an orally available PAD4 inhibitor preserves LV diastolic function and dampens disease progression in mice with collagen-induced arthritis (CIA).**

**(A, E)** Time course illustrating the ratio of early (E) to late (A) diastolic filling velocities across the mitral valve as measured by pulsed-wave Doppler echocardiography (E/A ratio; left Y axis; —) in mice with CIA treated with vehicle and mice with CIA treated with the PAD4 inhibitor (n = 8). Clinical severity of rheumatoid arthritis (RA score; right y axis; —) in DBA/1 J mice with CIA treated with vehicle and mice with CIA treated with the PAD4 inhibitor (n = 8). **(B)** Representative flow patterns, acquired using pulsed-wave Doppler echocardiography, depicting the velocities over the mitral valve at an indicated timepoint in a mouse with CIA treated with the vehicle and a mouse with CIA treated with the PAD4 inhibitor. **(C)** E/A ratio and ejection fraction (EF) as assessed by echocardiography in mice with CIA treated with the vehicle and mice with CIA treated with the PAD4 inhibitor, respectively at day 56 (n = 8). **(D)** Representative LV section images of Sirius Red fast Green Staining Kit and comparative analysis of total fibrotic tissue (red) in CIA and CIA treated with the PAD4 inhibitor, respectively (scale bar: 50 $\mu$m). Arrows indicate fibrotic tissue (n = 8). **(E)** Deceleration time reflecting the duration for equalizing the pressure difference between the left atrium (LA) and the left ventricle (LV) and isovolumetric relaxation time in mice with CIA treated with the vehicle and mice with CIA treated with the PAD4 inhibitor (n = 7). **(F)** Representative Immunofluorescence staining of LV sections from a mouse with CIA and a healthy control for CD31$^+$ cells (green) and collagen type 1 (red) (scale bar: 50 $\mu$m). Quantification of total collagen type 1 deposition per LV section (n = 5/6). **(G)** Plasma levels of brain natriuretic peptide, as measured by ELISA on day 56 in mice with CIA and a healthy control group. (n = 12) as determined by ELISA and heart weight normalized to tibia length of mice with CIA and mice with CIA treated with the PAD4 inhibitor after 56 d. (n = 8). Data are mean ± SEM. *$P$ < 0.05, **$P$ < 0.01, ***$P$ < 0.001; paired $t$ test. **(D, E, F)** Wilcoxon matched pairs signed rank test.

± 274.4 pg/ml; n = 12; $P$ = 0.032). This indicates, that the neutrophils are contributing to the heart IL-1$\beta$ production (Fig 3D).

### Thromboinflammation in and around the cardiac microvessels is present in mice with CIA

Next, we asked where the activated neutrophils predominantly reside and hence exert their detrimental functions. We found that activated (H3Cit$^+$) neutrophils in mice with CIA were three times more likely to be

located perivascularly than within the cardiac tissue (n = 5; $P$ = 0.02) (Fig 4E). At the same time, we evaluated VWF staining as a marker of endothelial activation. We had observed increased deposition of VWF in arthritic joints before and hypothesized that similar mechanisms could be driving neutrophil infiltration in the CIA heart as well (12, 17). Indeed, we found deposition of VWF in the cardiac vasculature of mice with CIA to be fivefold elevated when compared with healthy controls (n = 5; $P$ < 0.0001) (Fig 4A and B). Given the prothrombotic propensity of VWF, this increase could drive micro thrombosis in the CIA model and

as expected, mice with CIA showed two times more platelet aggregates per left ventricular section than healthy controls (n = 6; P = 0,03) (Fig 4C and D).

### Treatment with an orally available PAD4 inhibitor preserves LV diastolic function in mice with CIA

To evaluate the role of PAD4 in neutrophil activation and trafficking in CIA, we used a selective oral PAD4 inhibitor (JBI-589) administered once daily starting after the onset of clinical RA at day 30 for 26 consecutive days. Mice were divided into two groups with comparable arthritis scoring. One was to be gavaged with the PAD4 inhibitor, the other with the vehicle. We chose a once-daily administration as opposed to a twice-daily regime which significantly inhibits RA (3), in the hope that this regime would leave residual arthritis disease. Indeed, although mice treated with JBI-589 had lower arthritis scores in the early period of the model (Day 36 [9 ± 0.5 versus 5 ± 0.8; n = 8; P, 0.001], day 44 [12 (11–13) versus 9 (7–11); n = 8; P = 0.02]), there was no longer a difference in arthritis severity at the end point of the study at day 56 between treated and untreated mice (12 [±1] versus 12 [±0.9]; P = 0.5) (Figs 5 and S3C). Despite comparable disease severity, treatment with the PAD4 inhibitor was able to mitigate adverse myocardial remodeling throughout the treatment period. Mice with CIA treated with JBI-589 for 26 consecutive days had preserved diastolic function as reflected by the E/A ratio being comparable with healthy mice and 40% higher than the placebo CIA group (n = 8; P < 0.0001) (Fig 5A–C). Correspondingly, the cardiac cycle was intact in JBI-589-treated mice, exhibiting normal and hence shorter IVRT (15 [14–16] ms versus 22 [20–23] ms; n = 8; P = 0.008) and DT (21 [14–24] ms versus 30 [28–33] ms; n = 8; P = 0.04) than mice with CIA treated with vehicle. There was no difference in systolic left ventricular function between mice with CIA treated with vehicle and mice with CIA treated with the PAD4 inhibitor (n = 8; P = 0.09) (Fig 5C and E). Both groups had equal heart rates at the time of the measurement (n = 8; P = 0.1). Finally, treatment with JBI-589 significantly reduced plasma BNP levels in treated mice when compared with animals treated with the vehicle (106.8 ± 23.1 pg/ml versus 55.4 ± 14.0 pg/ml; n = 8; P < 0.0001) (Fig 5G). Histological analysis suggested that the functional improvement was likely because of the prevention of myocardial hypertrophy and myocardial fibrosis (Fig 5D and F). Mice treated with JBI-589 had lower heart weight-to-tibia length ratio (9 [±2] mg/mm versus 11 [±2] mg/mm; P = 0.004) and reduced LVM measured by echocardiography (33 [27–39] mg versus 74 [43–94] mg; n = 8; P = 0.03) compared with mice treated with vehicle (Fig 5G). Mice treated with the PAD4 inhibitor had less total myocardial fibrotic tissue (4 [3–4]% versus 10 [5–11]%; n = 7–8; P = 0.01) and five times lower collagen type 1 deposition in the LV then the vehicle treated control (n = 5; P = 0.02) (Fig 5F).

### Inhibition of PAD4 reduces thromboinflammation by decreasing neutrophil infiltration, neutrophil H3Cit expression, and endothelial activation

Consistent with our results showing a decrease in myocardial fibrosis and lack of hypertrophy, JBI-589-treated mice had reduced numbers of neutrophils per mg LV heart tissue, determined by FC

analysis (3 [2–9] cells/mg heart versus 10 [6–14] cells/mg heart; n = 8 P = 0.03). Immunofluorescence, respectively, showed decreased cell density of H3Cit+ and Ly6G+ double-positive cells in JBI-589-treated mice with CIA when compared with mice with CIA treated with the vehicle (7 [±7] cells/mm$^2$ versus 212 [±73] cells/mm$^2$; n = 7 P < 0.0001) (Fig 6A and B). There was no difference in the number of circulating neutrophils at any timepoint between the groups.

Consistent with our assumption that VWF could be involved in neutrophil recruitment in mice with CIA, mice with CIA treated with JBI-589 had lower levels of deposited VWF when compared with mice with CIA treated with the vehicle (0.3 [0.1–0.7]% versus 3 [3–4]%; n = 8; P = 0.003) (Fig 6C). Finally, mice with CIA treated with vehicle had significantly higher heart tissue levels of IL1-$\beta$ when compared with mice treated with the PAD4 inhibitor (938, 6 [±610] pg/ml versus 221.2 [±57.1] pg/ml; P = 0,009) (Fig 6D).

Thromboinflammation with elevated levels of inflammatory cytokines, release of NET biomarkers in plasma, and decrease in health parameters were key components in CIA (Fig S1A, B, D, and E). Treatment with JBI-589 as expected reduced NET formation (n = 4; P = 0.03) (Fig S3B) but also improved health parameters such as body weight (n = 8; P = 0,009) (Fig S3A). Simultaneously, treatment with JBI-589 reduced the expression of pro-fibrotic markers, whereas mice with CIA showed increased levels of the pro-fibrotic mediator TGF-$\beta$ in heart tissue (n = 5; P = 0.002) treatment with JBI-589 reduced deposition of TGF-$\beta$ considerably (n = 6; P = 0.003) (Fig S2A–C). The production of TGF-$\beta$ has been implicated in driving the pro-fibrotic shift in heart tissue through fibroblast to myofibroblast phenoconversion (22, 23). Correspondingly, we observed a fivefold increase in the expression of alpha-smooth-muscle actin signal, a marker of activated myofibroblast (n = 5; P = 0.04) (Fig S2D and E). Treatment with JBI-589, in turn, reduced myofibroblast concentration (n = 5; P = 0.02) and TGF-$\beta$ tissue levels indicating (Fig S2C), that PAD4 inhibition affects TGF-$\beta$ production (Fig S3D and E). Thus, inhibition of PAD4 does target both thromboinflammation and profibrotic remodeling.

## Discussion

In this study, we show that treatment with the new orally available PAD4 inhibitor JBI-589 is able to mitigate adverse myocardial remodeling in chronic inflammatory arthritis (3, 24). Of particular significance, JBI-589 administration after the onset of clinical arthritis decreased neutrophil recruitment to and deposition of NETs in the myocardium, and lowered cardiac levels of inflammatory and pro-fibrotic cytokines. This provides further evidence for an accentuated risk of collateral organ damage in chronic inflammatory arthritis and supports a prominent role of PAD4 in the underlying pathophysiological mechanism.

Aberrant activation of neutrophils through immune complex precipitation at the joints is characteristic for arthritis (25). It is of interest to deepen the understanding on how, in this way, localized chronic inflammation triggers collateral damage. There is increasing evidence implicating neutrophils as being cellular mediators by locally delivering bioactive molecules such as Il-1$\beta$ (26). Also, NETs derived from activated neutrophils may induce organ

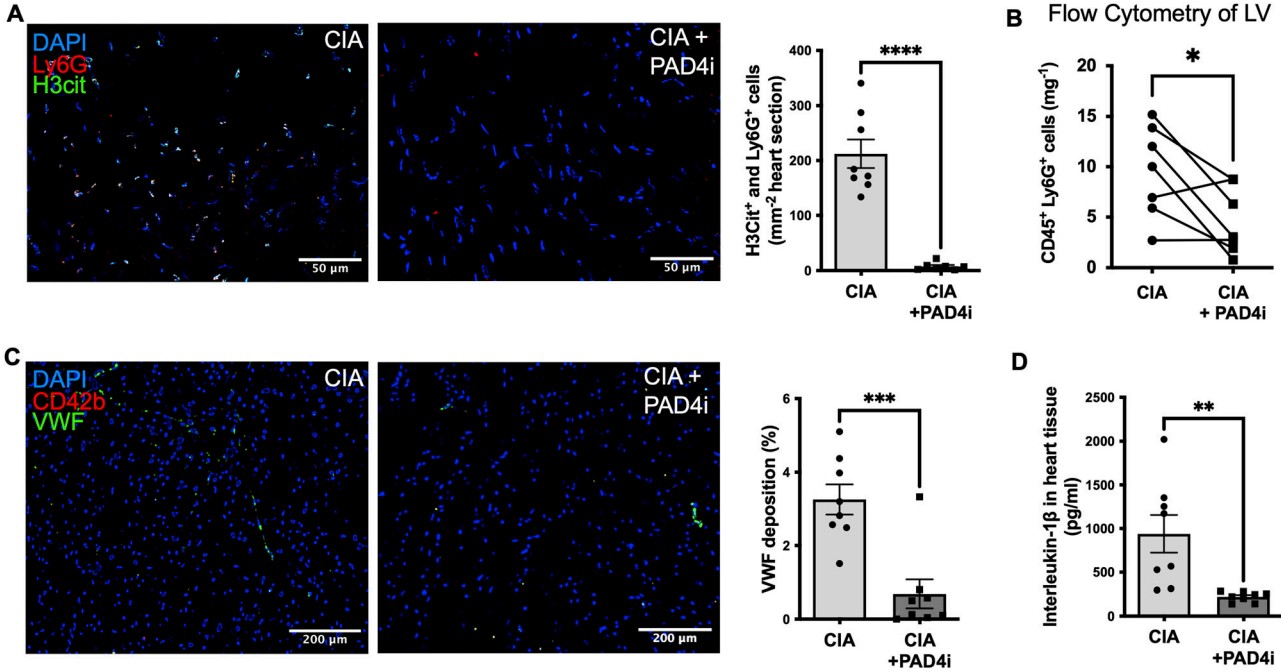

**Figure 6. Inhibition of PAD4 dampens thromboinflammation by decreased neutrophil infiltration, neutrophil H3Cit expression and reduced endothelial activation.**
**(A)** Representative immunostainings of LV sections and quantification of double-positive (Ly6G⁺ [red] and H3cit⁺ [green]) cells indicative of activated neutrophils in the myocardium of mice with collagen-induced arthritis (CIA) treated with the vehicle and mice with CIA treated with the PAD4 inhibitor, respectively (scale bar: 50 $\mu m$) (n = 7–8). **(B)** Quantification of infiltrating CD45⁺ and LY6G⁺ cells in the myocardial tissue of mice with CIA treated with the vehicle and mice with CIA treated with the PAD4 inhibitor, respectively, assessed by flow cytometry (n = 8). **(C)** Representative LV sections stained with DAPI⁺ (blue) and VWF⁺ (green). Quantification of VWF deposition in the myocardium of mice with CIA treated with the vehicle and mice with CIA treated with the PAD4 inhibitor (scale bar: 200 $\mu m$). **(D)** Myocardial tissue levels of IL-1$\beta$ in mice with CIA treated with the vehicle and mice with CIA treated with the PAD4 inhibitor. Data are mean ± SEM. *$P$ < 0.05, **$P$ < 0.01, ***$P$ < 0.001; unpaired $T$ test. **(B)** Wilcoxon matched pairs signed rank test.

damage distant from the original injury (27). Our results support the concept that PAD4, with its critical role in both the activation of neutrophils and extrusion of NETs, is involved in the mechanisms promoting collateral damage in arthritis, presumably through increased neutrophil recruitment and activation. There are several pathways through which PAD4 could influence neutrophil migration, both directly and indirectly. For one thing, being the only PAD equipped with a nuclear localization sequence, PAD4-mediated citrullination is considered to be an important factor in the transcriptional regulation of the expression of various genes (28). It is therefore of interest that PAD4 plays a prominent role in in the regulation of *CXCR2* (CXC-Motiv-Chemokinrezeptor 2) expression, a key cytokine receptor involved in neutrophil trafficking (29). Recent findings in cancer involving PAD4 in the formation of distant premetastatic niches with inhibition of PAD4 via JBI-589 blocking neutrophil recruitment into the primary tumor and metastasis also support this paradigme (24). Furthermore, we recently established the role of PAD4 in NLRP3 inflammasome formation/activation. Activation of the multiprotein complex NLRP3 inflammasome promotes the release of Il-1$\beta$, an important endothelial activator active in leukocyte recruitment (30, 31). It is therefore likely that inhibition of PAD4 also affects the cytokine milieu of treated animals and, consequently, neutrophil migration (15, 16).

What draws the activated neutrophils generated in the setting of arthritis to target the heart is not known. Alterations of the heart

vasculature can direct organ specificity to autoimmune attacks (32). We found increased VWF release/deposition in the myocardial vasculature of mice with arthritis and HF. In agreement, clinical data in humans demonstrates enhanced levels of endothelial adhesion molecules in the microvasculature of HF patients (33). Elevated PAD4-mediated neutrophil activation in arthritis could—in a vicious cycle—be a trigger for the release of additional VWF from Weibel Palade bodies facilitating the stabilization and endothelial anchorage of NET fibers fueling thromboinflammation (14, 34). Indeed, inhibiting PAD4 in our study resulted in reduced VWF surface expression in the myocardial vasculature. This could not only stem from reduced levels of PAD4-mediated NETosis in treated animals but from the role of extracellular PAD4 which, when released with NETs, reduces the activity of ADAMTS13, a plasma metalloprotease that cleaves VWF (12, 14).

There is evidence of elevated immune cell infiltration in the livers, kidneys, and lungs of arthritic mice (35, 36). Inhibition of PAD4 could therefore not only be beneficial to the primary disease but several collateral comorbidities of RA as well. With its multiple functions, both within different cell types and extracellularly, this detrimental effect of PAD4 is likely not limited to its influence on neutrophils and NETs alone (2, 37). However, its crucial role in amplifying the neutrophil inflammatory response closely links PAD4 and neutrophils in arthritis and arthritis-related HF (38).

We report extensive hypertrophic concentric myocardial remodeling, increased cardiomyocyte cell size, and increased fibrosis

with collagen type 1 deposition in the heart tissue of mice with arthritis. This resulted in clinically evident impaired diastolic left ventricular relaxation and stiffness and increased HF biomarker levels.

Activation of PAD4 is closely linked to adverse myocardial remodeling. We have shown, in aged mice and in an experimental model of cardiac fibrosis, that PAD4 deficiency protected hearts from fibrosis and ventricular remodeling (39). In ex vivo models, myofibroblasts treated with NETs demonstrate increased connective tissue growth factor expression, collagen production, and proliferation/migration (40). This is in line with our results showing that treatment with JBI-589 reduces myocardial fibrosis and collagen type 1 deposition in the myocardium, thus preventing adverse myocardial remodeling with resulting HF in arthritis.

It is difficult to quantify a potential contributing effect of CIA pathology in noncardiac tissues which might be an additional factor for the development of HF (i.e., kidney failure or pulmonary hypertension) (9). However, the fact that both treated and untreated arthritic mice had comparable clinical severity score in the end renders it reasonable to argue that PAD4 activity is prominently involved in HF development in this late stage of arthritis. We believe that this could be because of increased PAD4-mediated neutrophil activation and NETosis in mice with CIA in cardiac tissue. The subsequent release and anchorage of VWF by NETs on the endothelial cell surface promotes platelet recruitment and secretion of profibrotic cytokines such as TGF-$\beta$ (41, 42). TGF-$\beta$ in turn is critically involved in myofibroblast transdifferentiation driving cardiac fibrosis (43). This is in line with our results showing increased deposition of myofibroblasts and TGF-$\beta$ in the cardiac tissue of mice with CIA. Downstream from PAD4-mediated neutrophil activation, NET components such as myeloperoxidase and histones have independently been shown to promote fibrosis and a direct adverse effect of inflammatory cytokines such as IL-1$\beta$ on cardiomyocytes has been described possibly adding to the cardioprotective effect of PAD4 inhibition (44, 45). Our results show reduced levels of IL-1$\beta$ in the heart tissue of arthritic mice treated with JBI-589. This further supports a relationship between PAD4 and the NLRP3 inflammasome, the major source of IL-1$\beta$ (15). It is reasonable to suspect that local IL-1$\beta$ release in arthritis plays an integral role in attracting additional inflammatory cells and platelets. This again promotes cardiac release of TGF-$\beta$ (46).

Beyond that, NETs are recognized to attract functional tissue factor, containing microparticles, and to induce platelet activation and further thrombin generation propelling microvascular thrombosis (47).

The impaired LV relaxation pattern combined with an increase in BNP and the preserved LV ejection fraction (Fig 1) observed in CIA bears a striking resemblance to the HF subtype HFpEF, a critical public health problem increasing in prevalence (20, 48, 49, 50). The findings of this study support a key role of PAD4 and NETs in driving the pathophysiology of HFpEF that accompanies a variety of disorders with an inflammatory component and suggest CIA as an easy, accessible model to study HFpEF pathophysiology (51, 52). It is possible that an underlying chronic activation of neutrophils is a prominent etiological factor of HFpEF. However, this warrants further characterization of the model and a thorough examination taking into consideration the multiple sites of action/injury in CIA that could affect cardiac function outside neutrophil activation.

In conclusion, we provide evidence for a functional role of PAD4 and NETs in driving the inflammation-related HF in murine arthritis. Our data support the role of inflammation-exposed activated neutrophils in cardiac IL-1$\beta$ and TGF-$\beta$ release, and thus, secondary organ damage. Inhibition of PAD4 could be a promising target to limit both the primary disease and the resulting thromboinflammation driving collateral damage in autoimmune disease.

# Materials and Methods

### Animals

DBA/1 J mice were purchased from Jackson Laboratory. All mice were kept specific pathogen-free. Experimental protocols were approved by the Institutional Animal Care and Use Committee of Boston Children's Hospital (Protocol number: 20-01-4096R). To conduct a time-course study investigating onset and persistence of the cardiac dysfunction in DBA/1 J, the following groups were studied: DBA/1 J with CIA, DBA/1 J age-matched wild type, DBA/1 J with CIA treated with the PAD4 inhibitor, DBA/1 J with CIA treated with 0.5% Methylcellulose (vehicle of the PAD4 inhibitor). The PAD4-specific inhibitor JBI-589 was provided by Jubilant Therapeutics (JBI-589, (R)-(3-aminopiperidin-1-yl)(2-(1-(4-fluorobenzyl)-1H-indol-2-yl)-3-methylimidazo[1,2-a]pyridin-7-yl)methanone). Mice received JBI-589 at a dose of 10 mg/kg via oral gavage for 26 consecutive days. JBI-589 was given in suspension formulation prepared using Tween-80 and 0.5% methyl cellulose. For the PAD4 inhibitor trial, mice were paired depending on their RA severity score and randomly allocated to treatment or control group. All recurring events including animal scoring and administration of inhibitor was done at similar times during the day. All groups were age and sex matched and were fed ad libitum with free access to water. All procedures conformed to the NIH Guide for the Care and Use of Laboratory Animals. Experimental protocols were approved by the Institutional Animal Care and Use Committee of Boston Children's Hospital (Protocol number: 20-01-4096R).

### CIA model and scoring

DBA/1 J mice aged 8–9 wk were immunized with an emulsion of 50 $\mu$l CFA (Catalog no. 7023; Chondrex, Inc.) and 50 $\mu$l of bovine type II collagen (catalog no. 20012; Chondrex, Inc.) injected intradermally into the base of the tail on day 0. On day 21, a booster immunization of type II collagen with Freund's incomplete adjuvant (catalog no. 7002; Chondrex) was administered intradermally at a site proximal to the first injection site. Mice were assessed for development of arthritis using the semiquantitative, mouse arthritis scoring system provided by Chondrex (www.chondrex.com). This protocol is based on a hind-foot examination with range of 0 (no inflammation) to 4 (erythema and severe swelling encompassing ankle, foot, and digits). Evaluation of arthritis severity was performed blinded by two independent evaluators.

### Echocardiography

B-Mode, M-mode, and Doppler echocardiography were performed in DBA/1 J mice with CIA, DBA/1 J mice with CIA treated with PAD4 inhibitor or saline administration, and age-matched DBA/1 J mice at day 0, day 25, day 45, and day 56 for time-course analysis of cardiac function. Anesthesia was induced with 3% isoflurane and maintained at 1.5–2% for the duration of the procedure. Warmed echo gel was placed on the shaved chest. Body temperature was regulated through a heat pad and heart rate measured with an electrocardiogram; both were kept consistent between experimental groups (37°C and 400–500 bpm). Echocardiography images were recorded using a Vevo 3100 imaging system with a 25–55-MHz linear probe (MX550D; VisualSonics). Percentage of ejection fraction was calculated in the parasternal long-axis view. M-mode was measured at the papillary muscle level in the short-axis view. Parameters measured in M-mode/SAX include the following: fractional shortening (FS), LVPWd, left ventricular internal diameter in diastole. From these measured values, other estimates were calculated by a computerized algorithm of the echocardiograph. Vevo LAB ultrasound analysis software version 5.7.1; FUJIFILM VisualSonics was used for calculations such as LVM (LV Mass = 0.8 × (1.04 × ((LVEDD + IVSd +PWd)3 − LVEDD3)) + 0.6) or RWT (calculated as two times the LVPWd divided by the left ventricular diastolic diameter). Diastolic transmitral left ventricle (LV) inflow images were acquired from apical four-chamber views using pulsed-wave Doppler to calculate early (E) and late (atrial, A) peak-filling blood flow velocities, IVRT, and E-wave deceleration time (DT). Tissue Doppler imaging to measure tissue motion velocity from the mitral annulus was used to confirm Doppler measurements. The E/A ratio represents the ratio of E wave to A wave. The measurement probe was positioned at the tips of mitral valve leaflet in the mitral valve annulus with the ultrasound beam positioned parallel to the direction of blood flow. All measurements were obtained in triplicate and averaged.

### Quantification of collagen deposition and cardiomyocyte size

Mice were euthanized by cervical dislocation under deep anesthesia with isoflurane (3–4% in oxygen) after verification of sufficient analgesia by a lack of visible response to a footpad squeeze. Euthanasia was confirmed with monitoring the mouse for respirations for at least 1 min after cervical dislocation. Hearts were collected and residual blood cleared by retrograde perfusion with 5 ml ice-cold PBS buffer. Hearts were fixed in 4% paraformaldehyde solution for 24 h, specimens and afterwards dehydrated in alcohol, embedded in paraffin, and cut into 8-$\mu$m-thick serial slides. After deparaffinization through xyline and rehydration through different grades of alcohols to distilled water, sections were fixed in Bouin's solution (Sigma-Aldrich) over night. After that, sections were stained according to the Masson's trichrome Staining Kit (Sigma-Aldrich) manufacturers guidelines. In addition, a collagen-specific Sirius Red/Fast Green Collagen Staining Kit (catalog # 9046; Chondrex Inc) was performed according to the manufacturer's protocols. For quantification, photographs of heart sections were taken by bright-field microscopy and quantified by an investigator blinded to the identity of the samples using ImageJ. For quantification of perivascular fibrosis, five pictures of the interstitial area and perivascular area were chosen at random for quantification and the average used for comparative analysis.

For cryosectioning, tissue samples were snap frozen in O.C.T (product Code 4583; Tissue-Tek). WGA (Catalog number: W11261; Invitrogen), labeling glycoproteins of the cell membrane was used for cardiomyocyte cell size quantification. Heart sections were fixed in 4% paraformaldehyde solution for 15 min at 37°C. WGA conjugate (concentration 5 $\mu$g/ml) was applied followed by incubation for 30 min at 37°C. After washing, the sections were permeabilized with 0.1% Triton X and counterstained with Hoechst 33342 (1:10,000; Invitrogen). Cardiomyocyte area ($\mu m^2$) was calculated using ImageJ cross-sectional analyzer (https://imagej.net/plugins/cross-sectional-analyzer) of an average of at least 400 individual cells. Both collagen inoculation and tissue harvest were done in the morning to allow for comparable physiological environments.

### Immunostaining

Cryosections of 8-$\mu$m were fixed in 4% paraformaldehyde, permeabilized with 0.1% Triton, blocked in 3% BSA, and incubated overnight at 4°C with the following primary antibodies: anti-collagen I (ab21286; Abcam); anti-Ly6G (clone 1A8; BioLegend); anti-H3Cit (ab5103; Abcam); anti-CD31 (553370; BD Pharmingen); anti-VWF (P0226; DakoCytomation); anti-CD42b (M040-0; Emfret); anti-CD45 (ab154885; Abcam); anti-TGF $\beta$ (ab66043; Abcam); anti-alpha smooth muscle actin (ab5694; Abcam); and anti-CD68 (ab53444; Abcam). After washing, sections were stained with the respective Alexa Fluor-conjugated secondary antibodies (Alexa Fluor 488 donkey anti-rabbit [Cat. Nr. A21206; Invitrogen] IgG and Alexa Fluor 555 goat anti-rat [Cat. Nr. A21434; Invitrogen] IgG; Alexa Fluor 647 goat anti-rabbit [Cat. Nr. A21244; Invitrogen]) and counterstained with Hoechst 33342 (1:10,000; Catalog number: H3570; Invitrogen).

### Neutrophil isolation and NETosis assay

Peripheral neutrophils were isolated as previously described (5). Briefly, blood was drawn into 15 mM EDTA, 1% ultra-low endotoxin BSA (1:2 vol/vol). After removal of plasma, cells were resuspended in PBS and layered onto a discontinuous Percoll gradient (78%/69%/52% made isotonic with the addition of 10X PBS; Merck). Samples were centrifuged for 32 min at 1,500$g$ at RT, without breaks. After several washing steps, isolated neutrophils were resuspended in RPMI + 10 mM HEPES (Catalog number: 15630080; Gibco) and plated in a 96-well plate in a concentration of 10–15 × 10$^3$ neutrophils per well. Neutrophils were allowed to adhere to the glass bottom for 15 min at 37°C and 5% CO$_2$ after which they were and fixed with 4% paraformaldehyde followed again by permeabilization, blocking, and immunostaining with anti-H3Cit (Abcam) and counterstaining with Hoechst 33342 (1:10,000; Catalog number: H3570; Invitrogen).

### Flow cytometry

After euthanasia, the heart was immediately perfused with 5 ml ice-cold 0.05% EDTA in PBS. The perfusion needle was inserted into the

LV and the right atrium cut with a scissor. Paleness of the coronary arteries and cardiac veins was visually verified before cardiac excision. Apical sections, representative of LV tissue, were removed, then placed into ice-cold PBS and afterwards minced. Minced sections were incubated in a solution containing 1.5 mg/ml collagenase (Product Nr. C8051; Merck) for 45 min at 37°C. After filtration through a 70-$\mu$m cell strainer, the single-cell suspension was centrifuged and the pellet resuspended in DMEM (Catalog Number 11965092; Thermo Fisher Scientific) topped up with cell debris remover (Order Number 130-109-398; Milteny Biotec) and PBS followed by another centrifugation at 3,000$g$. This separated the mononuclear cells from cellular debris. After centrifugation, both the upper and interphase were discarded. After resuspension with FC buffer (0.1% BSA, 2 mM EDTA in PBS), the single-cell suspension was blocked with Fc block (anti-mouse CD16/CD32, 1:100 dilution; BioLegend, Cat. No. 101320; TrueStain Fcx), followed by washing and staining with anti-CD45-PE (cat. No. 553081; BD Pharmingen), anti-Ly6G-Pacific Blue (Cat. No. 101224; BioLegend), and viability staining (Cat. No. 65-0864-14; Invitrogen). The single-cell suspension was washed once more and spiked with Count-Bright Absolute Counting Beads (Cat. No. C36950; Invitrogen). Samples were run on a BD LSR Fortessa (BD Biosciences) using FC Diva software and analysed on FlowJo software. Once the doublets (by FSC-H versus FSC-A) and dead cells (live versus dead) were excluded, neutrophils were identified as $CD45^+$ $Ly6G^+$ cells and further quantified using beads, lot-specific bead concentration, and the manufacturer's calculation guidelines.

### Tissue lysate preparation

For tissue lysate, tissue was dissected, washed with PBS, and homogenized vigorously. Tissue was then added to RIPA Lysis and Extraction Buffer (Catalog number: 89900; Thermo Fisher Scientific) and incubated for 30 min at 37°C. Afterwards, the tissue suspension was sonicated for 2–5 min in rounds of 10 s at time at a power of 180 W. Sample was kept on ice throughout the whole process. Protein levels were determined using Bradford assay dye-based protein detection (Catalogue Number 23236; Thermo Fisher Scientific) according to the manufacturer's protocol.

### Peripheral blood and plasma analysis

Blood was collected from anesthetized mice via the retroorbital sinus into EDTA-coated capillary tubes and was analyzed by a Hemavet 950FS (Drew Scientific) for complete blood counts. Platelet-poor plasma was prepared immediately after blood collection by centrifuging anticoagulated whole blood for 5 min at 2,300$g$. Plasma supernatant was carefully removed and centrifuged again for 10 min at 16,100$g$ to remove any remaining blood cells. Plasma samples were immediately stored at −80°C until analysis.

### Determination of plasma/tissue protein levels

ELISA was performed for IL-1$\beta$ (ELISA MAX Deluxe Set Mouse IL-1$\beta$; Catalogue Number 432616; BioLegend), IL-6 (ELISA MAX Deluxe Set Mouse IL-6; Catalogue Number 431316; BioLegend), and BNP (RayBio Mouse/Rat Brain Natriuretic Peptide EIA Kit; Catalog #: EIAM-BNP,

EIAR-BNP; MyBioSource) to determine protein plasma or tissue lysate levels. Western blot analysis was performed using anti-H4cit (ab81797; Abcam) and VWF (P0226; DakoCytomation) antibody (17).

### Statistics

Values were tested for a Gaussian distribution using the D'Agostino-Pearson omnibus normality test with a 95% confidence level or Kolmogorov–Smirnov test. Continuous variables are presented as medians ± lower and upper quartiles if they followed a non-Gaussian distribution and as means ± SEM if they followed a Gaussian distribution. Non-normally distributed variables were tested using the Mann–Whitney $U$ test for unpaired analysis and the Wilcoxon matched-pairs side rank test for paired analysis. Normally distributed values were tested using unpaired or paired $T$ test. Differences between more than two groups were compared using Kruskal–Wallis test or ordinary one-way ANOVA, respectively. Sample size calculation for the comparative analysis of WT DBA1 versus DBA1 with CIA was estimated from a previous study using echocardiography data from C57BL/6 male yielding a n = 12 mice in each group for 95% power at the 0.05 level of significance to detect a difference in E/A ratio assuming a mean (SD) of 1.5 (0.2) for the control group and a mean (SD) of 1.2 (0.2) for the CIA group. All figures are presented as mean ± SEM. In all cases, $P < 0.05$ was considered statistically significant.

## Data Availability

The data that support the findings of this study are available from the corresponding author upon reasonable request.

## Supplementary Information

## Acknowledgements

This work was funded by a grant from the National Heart, Lung, and Blood Institute to DD Wagner (NIHR35OIA HL135765), a grant from the German Research Foundation to LA Heger (HE 8679/1-1:1) and a grant from the Society for Thrombosis and Haemostasis Research e.V. (GTH) to N Schommer. The authors thank Kristen Douthit and Ella Ziegler for careful editing and manuscript preparation and Isabell Heger for her critical comments on the article.

### Author Contributions

LA Heger: conceptualization, data curation, formal analysis, investigation, visualization, methodology, project administration, and writing—original draft, review, and editing.
N Schommer: conceptualization, data curation, formal analysis, investigation, methodology, and writing—original draft, review, and editing.

S Fukui: conceptualization, data curation, and investigation.

S Van Bruggen: conceptualization, data curation, and writing—original draft.

CE Sheehy: conceptualization, investigation, methodology, and writing—original draft, review, and editing.

L Chu: conceptualization, data curation, investigation, and writing—original draft, review, and editing.

S Rajagopal: conceptualization, resources, investigation, and writing—original draft, review, and editing.

D Sivanandhan: conceptualization, resources, and writing—original draft, review, and editing.

B Ewenstein: conceptualization and writing—original draft, review, and editing.

DD Wagner: conceptualization, resources, data curation, formal analysis, supervision, funding acquisition, investigation, methodology, project administration, and writing—original draft, review, and editing.

## Conflict of Interest Statement

The authors declare that they have no conflict of interest.

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
