## [Reviewer comments · Life Science Alliance]

Life Science Alliance

Inhibition of Protein Arginine Deiminase 4 Prevents Inflammation-Mediated Heart Failure in Arthritis

Lukas Heger, Nicolas Schommer, Shoichi Fukui, Stijn Van Bruggen, Casey Sheehy, Long Chu, Sridharan Rajagopal, Dhanalakshmi Sivanandhan, Bruce Ewenstein, and Denisa Wagner

DOI: <https://doi.org/10.26508/lsa.202302055>

Corresponding author(s): *Denisa Wagner, Boston Children's Hospital and Lukas Heger, Boston Children's Hospital*

Review Timeline:

Submission Date:	2023-03-23
Editorial Decision:	2023-05-24
Revision Received:	2023-06-10
Editorial Decision:	2023-06-29
Revision Received:	2023-07-13
Accepted:	2023-07-14

Scientific Editor: Novella Guidi

Transaction Report:

May 24, 2023

Re: Life Science Alliance manuscript #LSA-2023-02055-T

Dr. Denisa D. Wagner
Boston Children's Hospital
Program in Cellular and Molecular Medicine
1 Blackfan Circle
Boston, Massachusetts 02115

Dear Dr. Wagner,

Thank you for submitting your manuscript entitled "Inhibition of Protein Arginine Deiminase 4 Prevents Inflammation-Mediated Heart Failure in Murine Arthritis" to Life Science Alliance. The manuscript was assessed by expert reviewers, whose comments are appended to this letter. We invite you to submit a revised manuscript addressing the Reviewer comments.

Thank you for this interesting contribution to Life Science Alliance. We are looking forward to receiving your revised manuscript.

Sincerely,

B. MANUSCRIPT ORGANIZATION AND FORMATTING:

Reviewer #1 (Comments to the Authors (Required)):

1. This manuscript addresses the role of PAD4 in inflammation-related heart failure (HF) in arthritis using an orally available inhibitor (JBI-589) and collagen induced arthritis in DBA/1J mice. The study characterizes cardiac functions and the inflammation-related heart failure in CIA, to conclude that CIA is a viable model to study the inflammation-mediated pathogenesis of HF. Using the selective PAD4 inhibitor, they propose PAD4-mediated neutrophil infiltration and NETosis around the cardiac vasculature as the mechanisms of HF development during arthritis. The findings are of significance to increase the knowledge on inflammation-related HF pathogenesis in arthritis.
2. The data support the claims. The used methods and experiments seem sound for the purpose and number of biological replicates appropriate in most cases. The findings are put into the context of the current knowledge in the discussion and the study limitations are briefly discussed too.
3. Minor comments
 - 3a. Please edit the following sentence to become easier to understand: "Furthermore, mice with CIA and presented a delay in flow velocity decline in early diastole resulting in a prolonged E-wave deceleration time (DT) (29 {plus minus}5 ms vs. 19 {plus minus}4 ms; n=12; p< 0.0001) when compared to a healthy control"
 - 3b. If an abbreviation for flow cytometry is needed, please use something else than FACS, which stands for Fluorescence Activated Cell Sorting.
 - 3c. Please write out Col1, and Col 1, for consistency.
 - 3d. The text in the figures, including axis titles, could have a larger font for clarity.

Reviewer #2 (Comments to the Authors (Required)):

The authors present interesting new findings regarding RA-associated inflammatory mechanisms that contribute to myocardial diastolic dysfunction via neutrophil NETosis. The authors use of the CIA model demonstrates increased neutrophilia and myocardial infiltration which is accompanied by increased NETosis. Increased neutrophil recruitment and activation is observed with RA induction and is associated with decreased passive filling and E/A and increased deceleration time and IVRT, fibrosis, thromboinflammation, and myocyte hypertrophy. A novel oral PAD4 inhibitor, JBI589, used to test the role of NETs in neutrophil associated HFpEF pathophysiology, demonstrates NETs contribute to the overall diastolic dysfunction and related pathology.

While the reversal of all heart pathology with PAD4 inhibition is clear from the data provided, the discussion does little to address how inhibiting PAD4 and NETosis has such a profound effect on neutrophil recruitment to the myocardium. Several questions arise regarding other responses with JBI589 treatment. Were circulating neutrophil levels affected? Was increased granulopoiesis responsible for increased numbers or was lifespan affected? Is there some relationship between neutrophil NETosis activation and recruitment? What might be the factors activating neutrophils in CIA?

Please state the time of day procedures and collection were performed and neutrophils exhibit a circadian rhythm which has been shown to affect numbers and effect.

A larger area or X-section of the trichrome staining should be provided to better understand of the extent of fibrosis. From Figure 2c it looks like fibrosis is patchy. The collagen hybridizing peptide may be a more sensitive approach to measuring active fibrosis and remodeling.

How were NETing neutrophils identified in Figure 3?

NET biomarkers, dsDNA and Cit3Histone, should be considered as a complex which can be measured using a sandwich ELISA and can be further validated with a dsDNA - MPO complex ELISA

Dear scientific editors:

Thank you for your effort in directing the review of our manuscript entitled "Inhibition of Protein Arginine Deiminase 4 Prevents Inflammation-Mediated Heart Failure in Murine Arthritis". Below, you will find our replies to the reviewers point by point. Changes in the MS are marked yellow.

Replies to reviewer # 1

1-2. We thank the reviewer for his/her positive comment on the manuscript.

3. Minor comments:

3a. *"Please edit the following sentence to become easier to understand: "Furthermore, mice with CIA and presented a delay in flow velocity decline in early diastole resulting in a prolonged E-wave deceleration time (DT) (29 {plus minus}5 ms vs. 19 {plus minus}4 ms; n=12; p < 0.0001) when compared to a healthy control" "*

We thank the reviewer for this comment. The sentence now reads: "Similarly, the duration for equalizing the pressure difference between the left atrium and the left ventricle (E-wave deceleration time [DT]) was prolonged in mice with CIA when compared to the healthy control group (29 ±5 ms vs. 19 ±4 ms; n=12; p < 0.0001).

3b. *If an abbreviation for flow cytometry is needed, please use something else than FACS, which stands for Fluorescence Activated Cell Sorting.*

Thank you for noticing the error. We changed to Flow cytometry.

3c. *Please write out Col1, and Col 1, for consistency.*

We altered the MS accordingly and put collagen type 1.

3d. *The text in the figures, including axis titles, could have a larger font for clarity.*

Thanks for this comment. We increased the font in our figures.

We thank the reviewer for careful reading of the manuscript and helpful suggestions.

Replies to reviewer #2

We are happy that the reviewer found our findings interesting.

**While the reversal of all heart pathology with PAD4 inhibition is clear from the data provided, the discussion does little to address how inhibiting PAD4 and NETosis has such a profound effect on neutrophil recruitment to the myocardium. Several questions arise regarding other responses with JBI589 treatment. Were circulating neutrophil levels affected? Was increased granulopoiesis responsible for increased numbers or was lifespan affected? Is there some relationship between neutrophil NETosis activation and recruitment? What might be the factors activating neutrophils in CIA?*

We thank the reviewer for this excellent comment which implementing helped to considerably increase the quality of our manuscript.

We believe there are several pathways how PAD4 can influence neutrophil migration both directly and indirectly. Recently, it was shown in a murine cancer model, that PAD4 regulates the expression of the major chemokine receptor CXC motif chemokine receptor 2 (CXCR2) which is comprehensively involved in the migration of neutrophils to different tissues after stimulation. While we already discuss this paper in our discussion we now elaborate on a possible regulatory function of PAD4 on CXCR2 within the complex mechanism of neutrophil

migration. We also suspect, that the interaction of PAD4 and the NLRP3 inflammasome with its direct link to the endothelial activator il1 beta (effector enzyme of NLRP3 inflammasome) is crucial for neutrophil recruitment in arthritis and now elaborate on it in our manuscript.

Page 14/15:

“Our results support, that PAD4, with its critical role in both, the activation of neutrophils and extrusion of NETs, is involved in the mechanisms promoting collateral damage in arthritis presumably through increased neutrophil migration and activation. There are several pathways how PAD4 could influence neutrophil migration and activation both directly and indirectly. For one thing, being the only PAD equipped with a nuclear localization sequence, PAD4 mediated citrullination is considered to be an important factor in the transcriptional regulation of various gene-expressions.(Li, Wang et al. 2010) It is therefore incidental, that recent evidence supports a prominent role of PAD4 in the regulation of CXCR2 (CXC-Motiv-Chemokinrezeptor 2) expression a key cytokine receptor involved in neutrophil trafficking. (Metzemaekers, Gouwy et al. 2020) Our results are supported by recent findings in cancer involving PAD4 in the formation of distant premetastatic niches with inhibition of PAD4 via JBI-589 blocking neutrophil recruitment into the primary tumor and metastasis. (Deng, Lin et al. 2022) Furthermore, we recently established a role of PAD4 in NLRP3 inflammasome formation/activation. Activation of the multiprotein complex NLRP3 inflammasome promotes the release of Il-1 β , an important endothelial activator active in leukocyte recruitment. (Kelley, Jeltema et al. 2019, Pyrrillou, Burzynski et al. 2020) It is therefore likely, that inhibition of PAD4 also affects the cytokine milieu of treated animals and consequently neutrophil migration. (Munzer, Negro et al. 2021, Fukui, Fukui et al. 2022)”

**Several questions arise regarding other responses with JBI589 treatment. Were circulating neutrophil levels affected? Was increased granulopoiesis responsible for increased numbers or was lifespan affected? Is there some relationship between neutrophil NETosis activation and recruitment? What might be the factors activating neutrophils in CIA?*

Thanks for this excellent comment. There was no difference in circulating neutrophil levels and others have shown, that PAD4 deletion does not affect neutrophil granulopoiesis. (Cela, D., S. L. Knackstedt, S. Groves, C. M. Rice, J. T. W. Kwon, B. Mordmuller and B. Amulic (2022). "PAD4 controls chemoattractant production and neutrophil trafficking in malaria." J Leukoc Biol 111(6): 1235-1242.)

We now add a sentence to it in our result section “There was no difference in the number of circulating neutrophils at any timepoint between the groups.” (page 14)

We believe, and our data implicates, that while activated neutrophils (through release of Il1 beta) and NETs promote local tissue injury and inflammatory responses such as endothelial activation and leukocyte recruitment. This is supported by our data showing increased VWF expression in heart tissue of mice with CIA which is reduced in CIA mice treated with the PAD4 inhibitor. In CIA neutrophils are activated by the local inflammatory response in the joints and the developing autoimmune disease. We propose, that this translates into a systemic neutrophil activation. Why these neutrophils are particularly detrimental to the heart is not known but is also observed in human inflammatory arthritic disease.

Page 15

Aberrant activation of neutrophils through immune-complex-precipitation at the joints is characteristic for arthritis. (Fresneda Alarcon, McLaren et al. 2021)

**Please state the time of day procedures and collection were performed and neutrophils exhibit a circadian rhythm which has been shown to affect numbers and effect.*

Both Collagen inoculation and tissue harvest were done in the morning to allow for comparable physiological environment. This was now added to the method section.

**A larger area or X-section of the trichrome staining should be provided to better understand of the extent of fibrosis. From Figure 2c it looks like fibrosis is patchy. The collagen hybridizing peptide may be a more sensitive approach to measuring active fibrosis and remodeling.*

We thank the reviewer. We believe the “patchy” stems for the propensity to form fibrosis around blood vessel where NETs are likely to form. We now provide larger areas of the trichrome staining in our supplement and discuss this in our manuscript.

** How were NETing neutrophils identified in Figure 3?*

NETs in the sections were detected by staining for DNA, and H3Cit the biomarker of NET citrullination during their formation, as described before. Short: Elongated H3cit signal with NET morphology close to DAPI signal was counted as NET.

(Munzer, P., R. Negro, S. Fukui, L. di Meglio, K. Aymonnier, L. Chu, D. Cherpokova, S. Gutch, N. Sorvillo, L. Shi, V. G. Magupalli, A. N. R. Weber, R. E. Scharf, C. M. Waterman, H. Wu and D. D. Wagner (2021). "NLRP3 Inflammasome Assembly in Neutrophils Is Supported by PAD4 and Promotes NETosis Under Sterile Conditions." Front Immunol 12: 683803.)

**NET biomarkers, dsDNA and Cit3Histone, should be considered as a complex which can be measured using a sandwich ELISA and can be further validated with a dsDNA - MPO complex ELISA*

We thank the reviewer for this comment. We did not perform this ELISA but quantified dsDNA and H3cit individually as outlined in our supplemental section.

We thank the reviewer for careful thinking about our manuscript and we are confident, that the reviewer's comments helped to considerably improve its quality.

June 29, 2023

RE: Life Science Alliance Manuscript #LSA-2023-02055-TR

Dr. Denisa D. Wagner
Boston Children's Hospital
Program in Cellular and Molecular Medicine
1 Blackfan Circle
Boston, Massachusetts 02115

Dear Dr. Wagner,

Thank you for submitting your revised manuscript entitled "Inhibition of Protein Arginine Deiminase 4 Prevents Inflammation-Mediated Heart Failure in Arthritis". We would be happy to publish your paper in Life Science Alliance pending final revisions necessary to meet our formatting guidelines.

- Please upload all figure files as individual ones, including the supplementary figure files; all figure legends should only appear in the main manuscript file
- please add ORCID ID for the corresponding (and secondary corresponding) author--you should have received instructions on how to do so
- please add a Category for your manuscript in our system
- it is important to ensure that the titles in the system and on the manuscript file match
- please consult our manuscript preparation guidelines <https://www.life-science-alliance.org/manuscript-prep> and make sure your manuscript sections are in the correct order
- please add an Author Contributions section to your main manuscript text
- please use the [10 author names et al.] format in your references (i.e., limit the author names to the first 10)
- please add callouts for Figures 1D, 4B, 5B and D, S1A-E, S2A-E and S3A-D to your main manuscript text;

Figure checks:

- please indicate scale bar sizes in Legends for your figures

A. FINAL FILES:

B. MANUSCRIPT ORGANIZATION AND FORMATTING:

Sincerely,

Reviewer #2 (Comments to the Authors (Required)):

I would like to congratulate the authors for their efforts in this study and thank them for their revised manuscript.

July 14, 2023

RE: Life Science Alliance Manuscript #LSA-2023-02055-TRR

Dr. Denisa D. Wagner
Boston Children's Hospital
Program in Cellular and Molecular Medicine
1 Blackfan Circle
Boston, Massachusetts 02115

Dear Dr. Wagner,

Thank you for submitting your Research Article entitled "Inhibition of Protein Arginine Deiminase 4 Prevents Inflammation-Mediated Heart Failure in Arthritis". It is a pleasure to let you know that your manuscript is now accepted for publication in Life Science Alliance. Congratulations on this interesting work.

DISTRIBUTION OF MATERIALS:

Again, congratulations on a very nice paper. I hope you found the review process to be constructive and are pleased with how the manuscript was handled editorially. We look forward to future exciting submissions from your lab.

Sincerely,
